# Deep inference of seabird dives from GPS-only records: Performance and generalization properties

**Amédée Roy**[1,2]*, **Sophie Lanco Bertrand**[1], **Ronan Fablet**[2]

**1** Institut de Recherche pour le Développement (IRD), MARBEC (Univ. Montpellier, Ifremer, CNRS, IRD), Sète, France, **2** IMT Atlantique, UMR CNRS Lab-STICC, Brest, France

* amedee.roy@ird.fr

## Abstract

At-sea behaviour of seabirds have received significant attention in ecology over the last decades as it is a key process in the ecology and fate of these populations. It is also, through the position of top predator that these species often occupy, a relevant and integrative indicator of the dynamics of the marine ecosystems they rely on. Seabird trajectories are recorded through the deployment of GPS, and a variety of statistical approaches have been tested to infer probable behaviours from these location data. Recently, deep learning tools have shown promising results for the segmentation and classification of animal behaviour from trajectory data. Yet, these approaches have not been widely used and investigation is still needed to identify optimal network architecture and to demonstrate their generalization properties. From a database of about 300 foraging trajectories derived from GPS data deployed simultaneously with pressure sensors for the identification of dives, this work has benchmarked deep neural network architectures trained in a supervised manner for the prediction of dives from trajectory data. It first confirms that deep learning allows better dive prediction than usual methods such as Hidden Markov Models. It also demonstrates the generalization properties of the trained networks for inferring dives distribution for seabirds from other colonies and ecosystems. In particular, convolutional networks trained on Peruvian boobies from a specific colony show great ability to predict dives of boobies from other colonies and from distinct ecosystems. We further investigate accross-species generalization using a transfer learning strategy known as 'fine-tuning'. Starting from a convolutional network pre-trained on Guanay cormorant data reduced by two the size of the dataset needed to accurately predict dives in a tropical booby from Brazil. We believe that the networks trained in this study will provide relevant starting point for future fine-tuning works for seabird trajectory segmentation.

## Author summary

Over the last decades, the use of miniaturized electronic devices enabled the tracking of many wide-ranging animal species. The deployment of GPS has notably informed on

**Data Availability Statement:** Data, code and fitted models are available on github repository https://github.com/AmedeeRoy/BirdDL.

**Funding:** This work is a contribution to the TRIATLAS project (European Union's Horizon 2020

research and innovation program – grant agreement No. 817578), and to the Young Team IRD Programm (JEAI) for TABASCO project. RF was supported by LEFE program (LEFE MANU project IA-OAC), CNES (grant SWOT-DIEGO) and ANR Projects Melody and OceaniX. Fieldworks have been conducted thanks to the cooperative agreement between IRD, the Agence Nationale de la Recherche (ANR) project TOPINEME, and of the International Joint Laboratory LMI-DISCOH (to SLB). The funders had no role in study design, data collection and analysis, decision to publish, or preparation of the manuscript.

**Competing interests:** The authors have declared that no competing interests exist.

migratory, habitat and foraging strategies of numerous seabird species. A key challenge in movement ecology is to identify specific behavioural patterns (e.g. travelling, resting, foraging) through the observed movement data. In this work, we address the inference of seabird diving behaviour from GPS data using deep learning methods. We demonstrate the performance of deep networks to accurately identify movement patterns from GPS data over state-of-the-art tools, and we illustrate their great accross-species generalization properties (i.e. the ability to generalize prediction from one seabird species to aother). Our results further supports the relevance of deep learning schemes as 'ready-to-use' tools which could be used by ecologists to segmentate animal trajectories on new (small) datasets, including when these datasets do not include groundtruthed labelled data for a supervised training.

## Introduction

Marine top predators have received significant attention in marine ecology over the last decades [1]. They are known to use vast areas for feeding, thus requiring specific adaptive foraging strategies in order to localize their preys, especially in the pelagic environments which are highly variable [2]. They offer a unique perspective into ocean processes and dynamics, given that they can amplify information on the structure of the seascape across multiple spatio-temporal scales due to their relatively high mobility and longevity. Often considered as sentinels of the environmental variability and bio-indicators for ecosystem structure and dynamics, their study has been particularly contextualized into ecosystem-based management and conservation issues [3, 4].

Numerous studies have focused on the variability of seabirds' foraging strategies and in particular of dive distributions. Assessing consistency or shifts in foraging locations [5–7], and in the resource spatial partitioning [8, 9] provide indeed crucial information for understanding marine ecosystems. This has been particularly enabled by great technical advances in the miniaturization and autonomy of biologging devices [10, 11]. GPS loggers have been at the forefront of this breakthrough, and can now provide precise and accurate data on the foraging trajectories of many free-ranging species, such as seabirds [12, 13]. Detailed information on the diving behaviour has also been gained through the additional use of pressure sensors, such as Time Depth Recorders (TDR) devices [14–16]. Yet, for historical, financial and ethical reasons, the deployment of several sensors has not always been possible and a substantial amount of tracking datasets consist in GPS tracks only. The development of tools dedicated to animal trajectories segmentation (i.e. for dive identification) is therefore needed to extract more out of historical seabird foraging trajectories [17].

Among existing approaches to dive identification from GPS tracks, many individual-based studies aim to infer behavioral state directly by applying thresholds to various ecological metrics of movement data, such as speed, direction and tortuosity [18, 19]. A common example is the so-called First-Passage Time method (hereafter, FPT), which is defined as the time taken for an individual to cross a virtual circle of given radius [20–22]. Here foraging behaviour is assumed to occur when birds fly at very low speeds [23]. Statistical methods have also been used to predict diving behaviour with clustering schemes such as the Expectation Maximization binary clustering technique [24, 25] or using hidden Markov models (hereafter, HMM) typically with 2 or 3 distinct behavioural modes to explicit account for time-related priors [26–29]. More occasionally, supervised machine learning approaches such as artificial neural

networks, support vector machines and random forests have also been used [30, 31]. We may refer the reader to [32] for a more detailed review of these methods.

Recently, deep learning methods have been suggested to be a potentially useful tool for behavioural pattern segmentation [33]. Deep learning refers to a neural network with multiple layers of processing units [34]. By decomposing the data through these multiple layers, deep neural networks may learn complex features for representing the data with a high level of abstraction at multiple scales. The trajectory of an animal being the result of complex processes at multiple spatio-temporal scales [35], deep learning might be able to extract relevant representations of trajectories for performing tasks such as classification, segmentation or simulation. Deep learning has become the state-of-the-art framework for a wide range of problems in text, speech, audio and image processing and applications in ecology have mainly addressed image analysis and computer vision case-studies [36, 37]. Fewer studies have explored deep learning for animal trajectory data. Recurrent neural networks have been used for movement prediction [38, 39], and for the identification of representative movement patterns [40]. Very recently, an attention network has also been proposed for comparative analysis of animal trajectories [41]. Related to our study, a fully-connected network (hereafter, FCNet) has been used to predict seabirds' diving in European shags, common guillemots and razorbills [17]. With a very simple FCNet with 4 layers comprising hundreds of hidden nodes, this study demonstrated the improved accuracy of this approach over commonly-used behavioural classification methods. These promising results support new investigations to further explore the potential of deep learning schemes for movement ecology studies.

In particular, a central challenge in deep learning is to make algorithms that will not only perform well on the training data, but also on new datasets [42]. Generalization properties are indeed crucial for deep networks to tackle a wide range of problems. For example, it would be relevant to develop a neural network for the segmentation of behavioral patterns of certain species and whose characteristics are transferable to the analysis of the behavior of another species. Transfer learning refers to the fact of using knowledge that was gained from solving one problem and applying it to a new but related problem. For this purpose, a solution known as 'fine-tuning' consists in using a pre-trained model as the initialization of the training scheme rather than training a new model from scratch [43].

As in [17], this work addresses the inference of seabird diving behaviour from GPS data using Deep Learning methods. Besides, their FCNet architecture, we investigated Convolutional Neural Networks and U-Networks [44], which are state-of-the-art architectures for time series and image data processing and shall better account for the time structure of trajectory data. As case-studies, we considered two tropical seabird genus with distinct diving behaviour (Boobies vs Cormorants). The associated datasets comprised 297 foraging trips derived from GPS data deployed simultaneously with pressure sensors for the identification of dives. Our specific objectives were therefore (a) to confirm the performance of deep networks over state-of-the-art tools for dives identification, (b) to demonstrate generalization properties of trained network to predict dives of seabirds from other colonies and (c) to evaluate the benefits of a transfer learning strategy known as 'fine-tuning' for accross-species generalization.

## Materials and methods

### Ethic statement

Tracking data were obtained from electronic devices attached to Peruvian boobies and Guanay cormorants tagged at the Pescadores and Guañape Islands, Peru, from 2007 to 2013, and from masked boobies tagged at the Fernando de Noronha archipelago, Brazil, from 2017 to 2019. This work was conducted with the approval of the Peruvian federal agency, Programa de

Desarrollo Productivo Agrario Rural, commonly known as "Agrorural". Headquarters of Agrorural are located at Av. Salaverry 1388, Lima, Peru, and of the Brazilian Ministry of Environment—Instituto Chico Mendes de Conservação da Biodiversidade (Authorization No 52583-5).

## Dataset

GPS and TDR devices were jointly fitted to breeding tropical seabirds both in Peru (92 Peruvian boobies, 106 Guanay cormorants) and Brazil (37 masked boobies). Peruvian boobies (*Sula Variegata*) and Guanay cormorants (*Leucocarbo Bougainvilli*) were captured at Isla Pescadores (11˚46'30.34"S, 77˚'51.22"W) every year in December from 2008 to 2013 and at Isla Guanuape (8˚18.92"S, 78˚'42.72"W) in December 2007, while masked boobies (*Sula dactylatra*) were captured at Fernando de Noronha archipelago (3˚'9.71"S, 32˚'36.11"W) every year in April from 2017 to 2019. The GPS were attached with Tesa tape on the tail feathers for boobies and on the back feathers for cormorants for 1 to 2 days and the TDR were fixed on the bird's leg with a metal band. In total, GPS devices (Gipsy GPS, 25–30 g, Technosmart, Rome, Italy; i-gotU GPS GT 600, 25–30 g, Mobile action Technology, NewTaipei City, Taiwan; MiniGPSlog 30 g, Earth and Ocean GPS, Kiel, Germany; Axy-trek 14g, Technosmart, Rome, Italy) and time-depth recorders (TDRs, 3 g; resolution, 4 cm; G5 CEFAS Technology, Lowesoft, UK) were fitted to 235 seabirds.

After recovery, each GPS track was split into foraging trips by selecting locations further than a given distance to the colony and longer than a given time. Foraging trips were linearly interpolated to the TDR sampling resolution (i.e. 1s) and the coverage ratio was computed as in [17]. It is defined as the ratio between the number of recorded fixes and the number of fixes that should have been recorded with a perfectly regular sampling in a fixed temporal window. Amount of missing data is detailed in Table 1. True dives were defined by depth measured by TDR higher than 2 meters. Each GPS position was thus associated with a boolean value detailing the 'dive' status. This dataset consists therefore in a total of 297 foraging trips of seabirds with doubled-deployment GPS and TDR (see Table 1).

## Deep neural network architectures

In this work, we investigated deep neural networks. As baseline architecture, we considered the fully-connected network (FCNet) proposed in [17]. Besides, as described in Fig 1 we considered a fully convolutional neural network (CNNet) and a U-shape network particularly adapted to segmentation problems (UNet) [44]. We describe below these three architectures and the associated supervised training procedure. We refer the reader to [37] for an introduction to deep neural networks dedicated to ecologists.

**Table 1. Dataset overview.** General statistics on the four linearly-interpolated datasets used in this study. (m ± s) is for respectively mean and standard deviation.

| Species | Colony Location | Nb of trips | Trip Duration (min) | Dives (%) | Dives Duration (s) | Gaps (%) | Resting (%) |
|---------|-----------------|-------------|---------------------|-----------|--------------------|---------|-------------|
| *Sula variegata* | Pescadores Island, Peru | 132 | 64 ± 37 | 1.3% | 2.5 ± 1.3 | 2.2% | 4.4% |
| *Leucocarbo bougainvilli* | Pescadores Island, Peru | 79 | 143 ± 69 | 9.4% | 12.9 ± 14.1 | 25.5% | 36.6% |
| *Sula variegata* | Guañape Island, Peru | 22 | 162 ± 75 | 0.7% | 3.3 ± 2.5 | 1.5% | 6.6% |
| *Sula dactylatra* | Fernando de Noronha, Brazil | 64 | 491 ± 377 | 0.2% | 2.2 ± 1.4 | 6.1% | 33% |

Dives refer to the proportion of positions labeled as 'dive' (TDR-derived depth higher than 2 meters).

Gaps consist in the proportion of missing fixes that have been linearly interpolated.

Resting has been defined as the proportion of time with speeds inferior to 1 m.s-1 associated to non-diving behaviour.

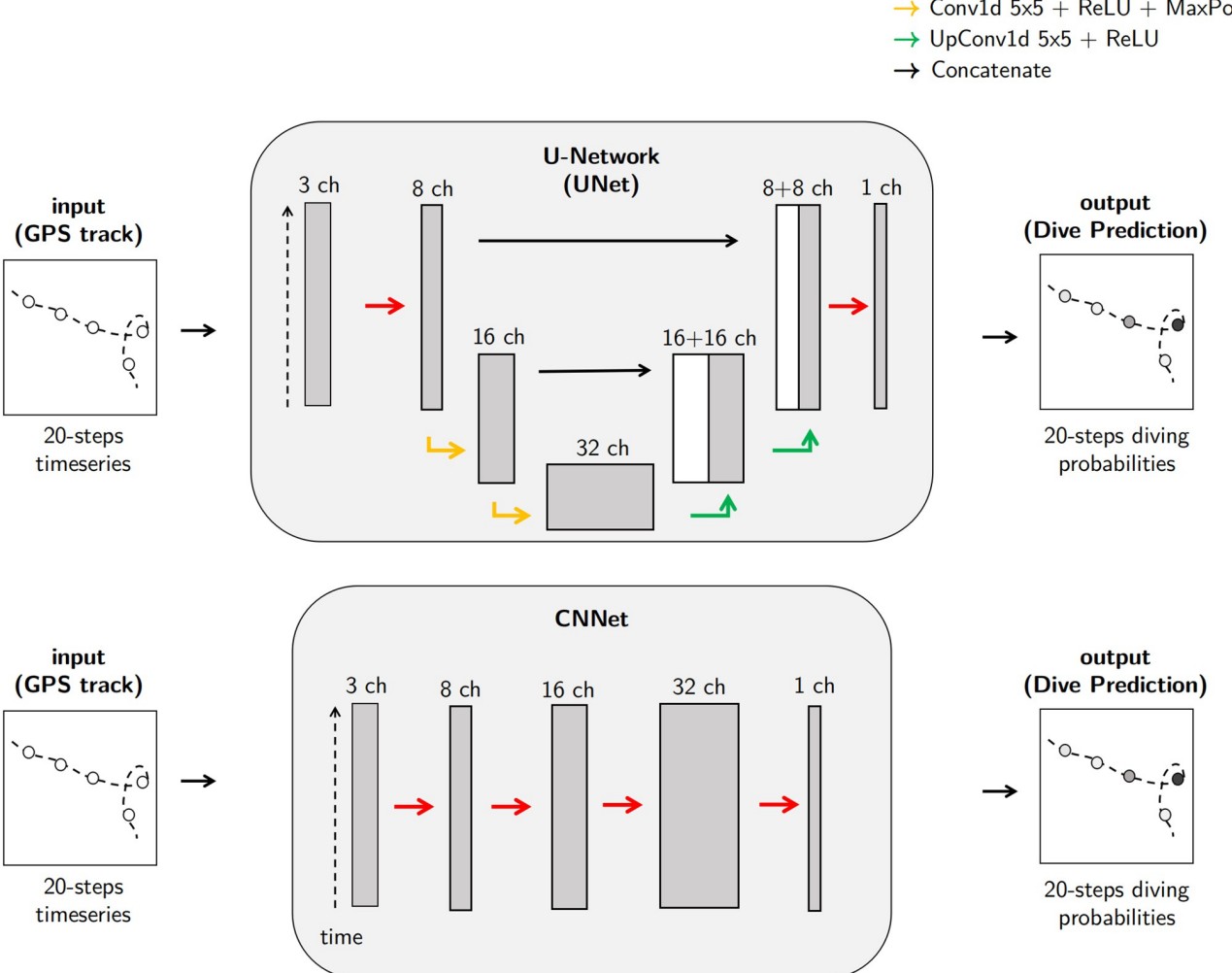

**Fig 1. Network architectures.** CNNet refers to a fully convolutional neural network. UNet refers to a U-shape network. A channel refer to deep learning terminology and describes a representation of the input data as output of some computation layer. Conv1d, MaxPool, and UpConv1d are abbreviations for usual deep learning operations. Details can be found on pytorch's documentation [45].

**Fully-connected network (FCNet).** The first architecture implemented was similar to the fully-connected network presented by [17]. As input vector, we used the concatenation of step speed, turning angle and coverage time series for over a 20-second window. This input vector is fed to a layer of 100 nodes followed by 3 layers of 500 nodes. Each node applies a linear transformation to the incoming data and a non-linear activation chosen as a Rectified Linear Unit (ReLU – $\text{ReLU}(x) = \max(0, x)$). The last layer applied a softmax binary function so that the output of the vector is a time series of values between 0 and 1, which can be interpreted as binary classification probabilities. This architecture is a classical example of a so-called multi-layer perceptron, with a rectified linear activation which is the default activation in deep learning architectures. Overall, this architecture involves 500k parameters.

**Fully convolutional networks (CNNet).** Convolutional networks exploit convolutional layers and are the state-of-art architectures for a wide range of applications, especially for signal and image processing tasks [46]. Thus, we investigated a basic neural network fully composed

of convolutional layers. Similar to FCNet, its input vector is the concatenation of step speed, turning angle and coverage time series over a 20-second window but its output is a vector of diving probability of the same length. Overall, this architecture CNNet involves 5k parameters.

**U-Network (UNet).**   As the considered problem can be seen as a segmentation task, a U-Net architecture naturally arises as a state-of-the-art solution [44]. The key feature of this architecture is to combine the information extracted by convolutional blocks applied at different temporal scales. To achieve this multi-scale analysis, the U-Net applies pooling layer to coarsen the time resolution and interpolation layers (UpConv1d layers) to increase the time resolution as sketched in Fig 1. At each scale, we apply a specific convolution block. We concatenate its output with the interpolated output of the coarser scale to a convolutional block, whose output is interpolated to the finer resolution. Overall, we may notice that the output of the U-Net architecture is a time series with the same time resolution as the input time series. Similarly to FCNet and CNNet, the last layer applies a sigmoid activation to transform the output into a time series of diving probabilities. Overall, this architecture U-Net involves 20k parameters.

**Network training and validation.**   Given a selected neural network architecture, the training procedure relies on a supervised learning scheme using a weighted binary cross entropy as loss function. This function evaluates the performance of a prediction by comparing the dive prediction (output of the model) with the true dives defined by TDR data. We consider a weighted version of the binary cross entropy because of the unbalanced presence of dive and no-dive behaviour in the studied trajectories (see Table 1). The objective is to penalize more for mistakes on the smaller class (diving behaviour) than for false positive, thus ensuring for convergence. In the reported experiments, the weight was empirically set to 5 for cormorants datasets and 30 for boobies dataset.

The minimisation of the training loss exploits the Adam stochastic optimizer [47]. A fixed learning rate of 0.001 was used for all training procedures. Networks were evaluated on training and validation datasets every epoch (defined as one pass through the entire train dataset). We consider an early-stopping criterion such that the training procedure was stopped as soon as the validation loss started increasing. Overall, given a trajectory the diving probability at a given location was assessed by computing the mean probability of all predictions derived from all 20 positions windows. These models were implemented, trained and tested with python using pytorch library [45]. Our pytorch Code is available on our github repository: https://github.com/AmedeeRoy/BirdDL.

## Benchmarked methods

Two classical methods for dive prediction First-Passage Time (FPT), and Hidden Markov Models (HMM) were evaluated for intercomparison purposes. FPT was computed following [48], by selecting the radius that maximizes the variance of passage times. Time passage values were converted into a probability of dives with min-max normalization. Regarding HMMs, we applied the momentuHMM R package [28]. We implemented HMMs with 3 (resp. 4) behavioural modes for boobies (resp. cormorants) associated to traveling, searching, diving and resting behaviours. This approach represents trajectories as a sequence of steps and angles. It models steps as random variables following a gamma marginal distribution and angle following a von mises marginal distribution. We may point out that the HMMs directly provide as outputs a likelihood value of the diving behaviour.

## Evaluation scheme

We describe below the evaluation scheme we implemented to assess the performance of the proposed neural network approaches. We first focus on the benchmarking of the performance

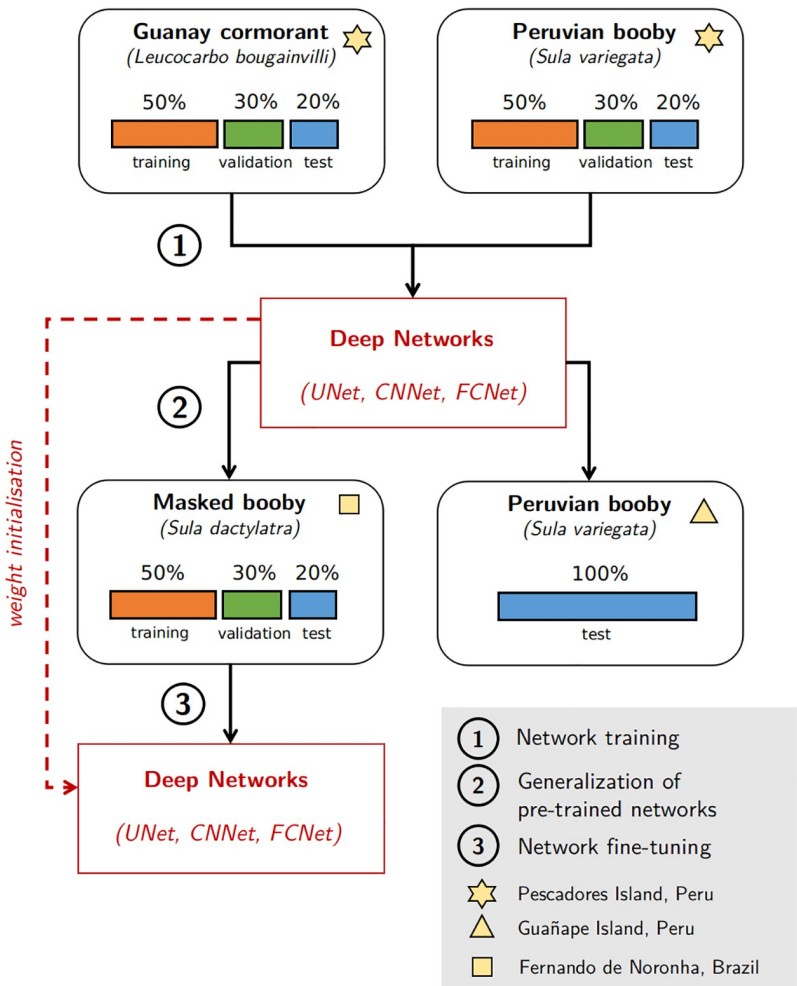

**Fig 2. Evaluation scheme.** (1) The datasets from Pescadores Island (see Table 1) have been used to train, validate and test deep networks. UNet, CNNet and FCNet refer to the deep network architectures used in this study (see Fig 1). (2) The trained networks have been directly used to predict dives on two other datasets without any additional training (Datasets from Guañape Island and Fernando de Noronha). (3) The dataset from Brazil have been used to train, validate and test deep networks. However, the deep networks previously obtained at step (1) have been used for weight initialization. This is known as Fine-tuning.

of the considered approaches in terms of dive prediction accuracy for different data input. For the proposed neural network architectures, we further analyze their generalization properties. The methodological framework is exposed in Fig 2.

**(a) Network training**. We assessed the dive prediction performance of the 5 benchmarked methods (FPT, HMM, FCNet, CNNet and UNet) considering trajectory data derived from the two dataset from Pescadores Island (see Table 1). To test for the effect of temporal resolution, the two datasets have been downsampled every 5, 15 and 30s. When downsampling, temporal windows containing at least one dive were classified as dives. Each dataset were then splitted into training, validation and test datasets with respective size of 50%, 30% and 20%. Deep networks were trained and selected based on the training and validation datasets. All approaches were finally compared on the testing dataset. Overall, this led to the quantitative comparison of the performance of 5 models on 6 datasets all listed in Table 2.

As evaluation metrics for dive prediction, we evaluated the receiver operating characteristics curve (ROC) which describes the performance of a binary classifier. It consists in plotting the true positive rate (i.e. true predicted dives) against the false positive rate (i.e. false predicted dives). We obtain this curve by varying the probability threshold defining dive/no dive behaviours. Moreover, we evaluate the area under the curve (AUC) as well as the binary cross entropy (BCE) for the test datasets. Regarding the AUC, it was estimated by integrating the ROC curve along the x axis using the composite trapezoidal rule. For neural network approaches, we also analyzed the value of the training loss for the training and test datasets.

**(b) Generalization performance of pre-trained networks**. In this section, we evaluated the generalization performance through the application of models trained on Pescadores dataset to data that have not been used during the training process. In particular, we evaluate previously fitted deep networks performance on the two datasets from Guañape Island and from Fernando de Noronha Archipelago, composed of Peruvian boobies and masked boobies trajectories, respectively. In this experiment, we compared the dive prediction performance of FPT and HMM methods to the best FCNet, CNNet and UNet models. Beyond AUC and BCE performance metrics, we also evaluated the relevance of the estimated maps of dive distributions. The later were computed using a weighted Kernel Density Estimator (KDE) using dive probabilities as weighing factor. As groundtruth, we considered the map of dive distributions estimated from true dive locations defined by TDR data. Kernel densities were estimated using a 0.01×0.01° grid and a bandwith of 0.25°. From these maps, we evaluated an means square error (MSE) as an integrated performance metrics for the different approaches [49].

**(c) Network fine-tuning**. We evaluated the benefits of fine-tuning for predicting dives of the 15s-resampled Brazilian dataset (Table 1) based on the deep networks fitted on the dataset from Pescadores. The Brazilian dataset was therefore split into training, validation and test datasets with respective size of 50%, 30% and 20%. We then trained models from scratch and using fine-tuning for the three studied network architectures and following the learning procedure presented before. We also evaluate the impact of the training dataset size by randomly selecting respectively 1, 5, 15 and 30 foraging trips for the training step. All models were finally compared to HMM and FPT methods on the testing dataset and using the AUC evaluation metric.

## Results

We detail below the numerical experiments performed in this study to assess the relevance of the proposed neural network approaches to predict dive behaviour of boobies and cormorants from trajectory data.

**(a) Network training**. On the Pescadores Island dataset used for network training, we reported a contrasted performance of the different methods, with AUC going from 0.61 to 0.96 (see Table 2), which corresponds in the best cases to correct prediction rates of diving and non-diving behaviour of approximately 95% and of 60% in the worst cases (see Fig 3). Overall, all methods performed better at predicting the dives of boobies than those of cormorants. The UNet obtained systematically the best prediction performance with averaged AUC of 0.93 (resp. 0.90) for boobies and cormorants respectively. The CNNet also achieved very good predictions, consistently performing at least as well as state-of-the-art tools with averaged AUC of 0.9 (resp. 0.85). The lowest performance was reported for the FPT approach, which never predicted dives with AUC higher than 0.73. The HMM obtained relatively good performance on the boobies dataset with AUC indices around 0.85, yet it did not get AUC higher than 0.76 on the cormorants dataset. It also obtained the highest BCE, approximately 2 to 10 times higher than the UNet. Regarding the FCNet, the AUC index ranged from 0.65 to 0.89, showing a

**Table 2. Deep networks training.** Performance metrics for all trained deep networks on the trajectories of Pescadores along with benchmarked methods used for comparison.

| Dataset | Resolution | Model | AUC | BCE | F-score | Train Loss | Validation Loss | Reference Name |
|---|---|---|---|---|---|---|---|---|
| SV (Pescadores) | 5s | FPT | 0.62 | 0.70 | 0.55 | - | - | - |
| | | HMM | 0.86 | 1.07 | 0.69 | - | - | - |
| | | FCNet | 0.89 | 0.38 | 0.81 | 0.61 | 0.61 | SV_FCNet_5s |
| | | CNNet | 0.94 | 0.29 | 0.88 | 0.48 | 0.49 | SV_CNNet_5s |
| | | **UNet** | **0.96** | **0.23** | **0.91** | **0.48** | **0.45** | **SV_UNet_5s** |
| | 15s | FPT | 0.71 | 0.79 | 0.66 | - | - | - |
| | | HMM | 0.87 | 2.39 | 0.84 | - | - | - |
| | | FCNet | 0.82 | 0.81 | 0.80 | 1.35 | 1.16 | SV_FCNet_15s |
| | | CNNet | 0.91 | 0.58 | 0.85 | 0.89 | 0.86 | SV_CNNet_15s |
| | | **UNet** | **0.93** | **0.57** | **0.86** | **0.87** | **0.79** | **SV_UNet_15s** |
| | 30s | FPT | 0.73 | 0.97 | 0.70 | - | - | - |
| | | HMM | 0.84 | 1.22 | 0.68 | - | - | - |
| | | FCNet | 0.82 | 1.10 | 0.79 | 1.69 | 1.74 | SV_FCNet_30s |
| | | CNNet | 0.85 | 0.98 | 0.80 | 1.55 | 1.47 | SV_CNNet_30s |
| | | **UNet** | **0.91** | **0.73** | **0.86** | **1.12** | **1.10** | **SV_UNet_30s** |
| LB (Pescadores) | 5s | FPT | 0.61 | 1.59 | 0.57 | - | - | - |
| | | HMM | 0.78 | 1.42 | 0.72 | - | - | - |
| | | FCNet | 0.87 | 0.40 | 0.79 | 0.55 | 0.67 | LB_FCNet_5s |
| | | CNNet | 0.92 | 0.30 | 0.84 | 0.48 | 0.57 | LB_CNNet_5s |
| | | **UNet** | **0.93** | **0.28** | **0.85** | **0.46** | **0.54** | **LB_UNet_5s** |
| | 15s | FPT | 0.58 | 1.73 | 0.62 | - | - | - |
| | | HMM | 0.76 | 3.35 | 0.72 | - | - | - |
| | | FCNet | 0.67 | 0.77 | 0.75 | 0.85 | 0.94 | LB_FCNet_15s |
| | | CNNet | 0.89 | 0.43 | 0.85 | 0.60 | 0.73 | LB_CNNet_15s |
| | | **UNet** | **0.90** | **0.37** | **0.83** | **0.52** | **0.76** | **LB_UNet_15s** |
| | 30s | FPT | 0.56 | 1.81 | 0.62 | - | - | - |
| | | HMM | 0.75 | 2.90 | 0.74 | - | - | - |
| | | FCNet | 0.65 | 0.92 | 0.75 | 0.96 | 1.10 | LB_FCNet_30s |
| | | CNNet | 0.74 | 0.74 | 0.76 | 0.86 | 1.01 | LB_CNNet_30s |
| | | **UNet** | **0.88** | **0.37** | **0.83** | **0.97** | **1.05** | **LB_UNet_30s** |

AUC means the Area Under the ROC curve. BCE is for binary cross entropy computed on the testing trajectories. Train and Validation Loss correspond to the loss computed after model training on respectively training and validation datasets. SV is for Peruvian boobies (*Sula variegata*), LB is for Guanay cormorants (*Leucocarbo bougainvilli*)

much greater variability than for CNNet and UNet architectures. The best neural network predictions for Pescadores dataset are illustrated in Fig 4. Interestingly, deep networks were relatively sensitive to the sampling resolution, whereas it did not affect much the performance of the FPT and HMM approaches. For both species, the higher the resolution, the better the performance for UNets, CNNets and FCNets. For instance, we reported a mean AUC of 0.92 for a 5s resolution vs. 0.81 for a 30s resolution. This was particularly true for the CNNet and mostly for the FCNet which did not performed better than HMM on the 30s-resoluted datasets, whereas they were able to outperform state-of-the-art approaches on the 5s-resoluted datasets.

**(b) Generalization performance of pre-trained networks**. Overall, all networks trained with data from Pescadores reported a AUC performance higher than 0.78 (resp. 0.56) when

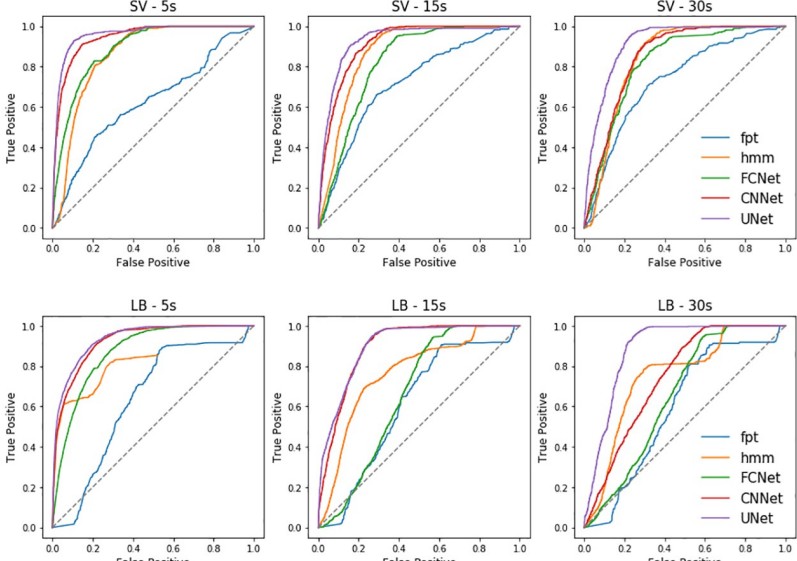

**Fig 3. Performance of deep networks on pescadores dataset.** ROC curves obtained from the prediction of 5 algorithms, First-Time Passage (FPT), Hidden Markov Models (HMM), Fully-Connected Network (FCNet), Fully-Convolutional Network (CNNet) and U-Network (UNet) on 2 distinct test datasets resampled at 3 different resolutions (5, 15 and 30s) derived from two seabirds species breeding in Pescadores Island from 2008 to 2013. SV stands for Peruvian boobies (*Sula variegata*), and LB stands for Guanay cormorants (*Leucocarbo bougainvilli*).

applied to Guañape (resp. Fernando de Noronha) dataset (Fig 5). AUC performance averaged 0.85 when using deep networks trained with the boobies dataset and 0.72 with cormorants data (see Table 3). On both datasets, the best models were UNet and CNNet models trained from boobies data with respectively AUC scores of 0.98 and 0.87. In particular, they outperformed the HMM that were specifically fitted to Guañape and Fernando de Noronha data. By contrast, the FCNet used by [17] that obtained better results than HMM on the Pescadores dataset (AUC of 0.89 vs 0.86 for HMM) did not predict better than HMM when used at Guañape (e.g. AUC of 0.89 vs 0.92 for HMM). The MSE for the estimated dive distribution maps stressed the greater relevance of UNet predictions with a MSE value 1.6 times smaller than the one derived from CNNet estimations and 1.9 times smaller than the one derived from HMM estimations (Table 3). As illustrated in Fig 6, only the Unet did not overestimate the number of dives in the vicinity of the colony as well as an other foraging area southward from the colony.

(c) **Network fine-tuning**. In this section, we evaluated the benefits of a fine-tuning strategy for the prediction of masked boobies dives. As expected, all deep networks initialized using previous models converged more quickly than deep networks trained from scratch. In particular, a dataset of 15 foraging trips (i.e. around 30k GPS positions) was enough for convolutional networks to obtain AUC of 0.9 using fine-tuning, whereas deep networks trained from scratch needed twice as many trips for the same predictive performance (Fig 7). The improvement issued from a fine-tuning was notably important for small-to-medium datasets (5-10 foraging trips, i.e. 10k to 20k GPS positions), and for the CNNet. It decreased as the size of the dataset increased. From our experiments, at least 5 trips were necessary to fine-tune a relevant network compared with the HMM baseline (see for instance bottom-center in Fig 7). The best neural network predictions for the Brazilian dataset are illustrated in Fig 4.

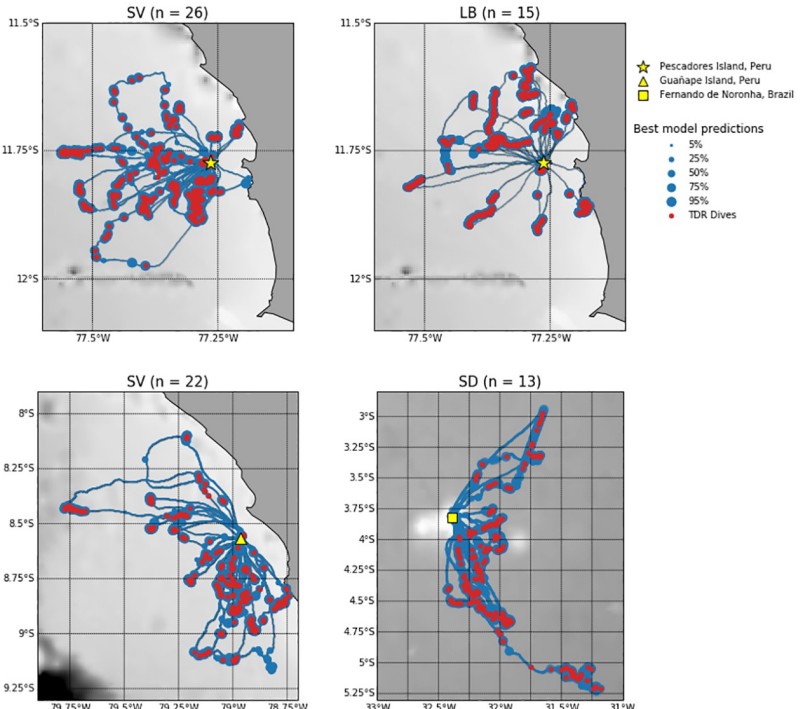

**Fig 4. Maps of predicted dives for all 'test' datasets.** Red points represent true dive derived from TDR data. Blue points represent diving probabilities at each location with radius increasing for higher probabilities. These probabilities are the outputs of the best deep networks for each dataset: Peruvian boobies from Pescadores (top left), and from Guañape Island (bottom left), Guanay cormorants from Pescadores Island (top right), and masked boobies from Fernando de Noronha archipelago (bottom right). SV stands for Peruvian boobies (*Sula variegata*), LB for Guanay cormorants (*Leucocarbo bougainvilli*) and SD for masked boobies (*Sula dactylatra*). Bathymetry is shown in grey and is extracted from GEBCO gridded dataset (https://www.gebco.net/). Land-sea mask is extracted from GSHHG data (https://www.soest.hawaii.edu/pwessel/gshhg/).

## Discussion

This study aimed at predicting seabirds dives from GPS data only using deep neural networks trained in a supervised manner based on TDR data to define the groundtruthed dives. In line with [17], this study further supports the relevance of deep learning approach over classical methods for dive predictions. Using convolutional architectures rather fully-connected ones, we reported even better results with higher stability to the different data inputs, as well as better generalization abilities.

Peruvian boobies and Guanay cormorants tracked in Peru breed in a highly productive upwelling system, the Humboldt Current System (HCS) and feed on the same preys, i.e. Peruvian anchovies [50]. However, they are known to have distinct foraging strategies: boobies are plunge divers reaching in average about 2 m depth and spending most of the time in fly, while cormorants dive deeper and longer on average, reach up to 30 m depth, and spend up to 40% of the time resting on the water surface [51] (Table 1). By contrast masked boobies breeding at Fernando de Noronha are plunge divers similarly to Peruvian boobies, yet they forage mainly in oligotrophic waters [52] and feed mainly on flying fish and flying squids [53, 54]. Their foraging strategies then differ from Peruvian boobies as they perform longest trips and spend more time resting at sea surface (Table 1). We demonstrated that for these three species, our best deep network models were able to accurately predict around 95% of dives and

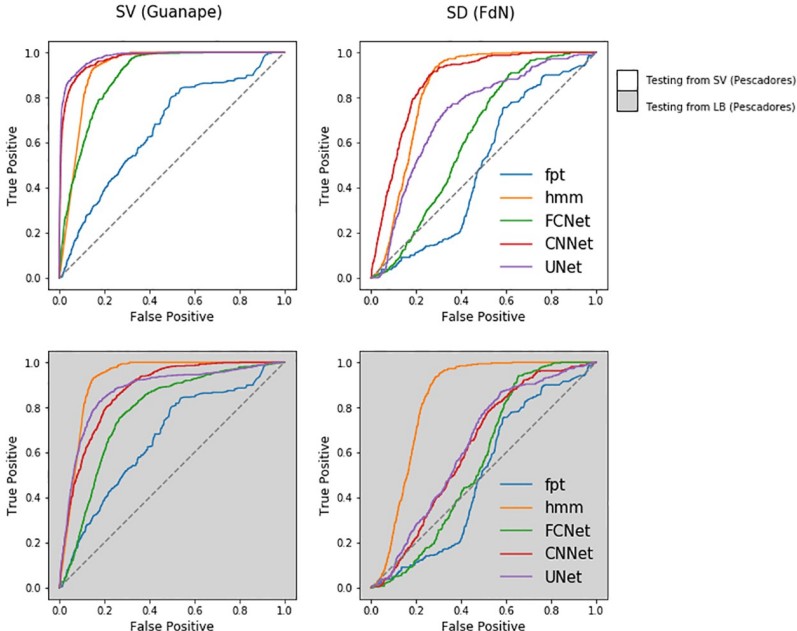

**Fig 5. Performance of tested deep networks.** ROC curves obtained from the prediction of 5 algorithms, First-Time Passage (FPT), Hidden Markov Models (HMM), Fully-Connected Network (FCNet), Fully-Convolutional Network (CNNet) and U-Network (UNet) on 2 distinct test datasets. SV stands for Peruvian boobies (*Sula variegata*), and LB stands for Guanay cormorants (*Leucocarbo bougainvilli*). The deep networks used in this figure have been trained on the dataset from Pescadores (see Fig 3 and Table 2). They have been used for dive prediction of Peruvian boobies in Guañape (left column) and for masked boobies in Fernando de Noronha (right column).

**Table 3. Deep network testing.** The deep networks fitted on the dataset from Pescadores have been used for dive prediction in Guañape and in Fernando de Noronha. Deep networks are described by their reference name (see Table 2).

| Dataset | Resolution | Model | AUC | BCE | F-score | MSE |
|---|---|---|---|---|---|---|
| SV (Guañape) | 5s | FPT | 0.65 | 0.57 | 0.43 | 10.9 |
| | | HMM | 0.92 | 2.46 | 0.88 | 7.7 |
| | | SV_FCNet_5s | 0.89 | 0.31 | 0.80 | 7.7 |
| | | SV_CNNet_5s | 0.97 | 0.20 | 0.91 | 6.5 |
| | | **SV_UNet_5s** | **0.98** | **0.10** | **0.91** | **4.0** |
| | | LB_FCNet_5s | 0.78 | 0.09 | 0.05 | 13.8 |
| | | LB_CNNet_5s | 0.87 | 0.07 | 0.07 | 7.8 |
| | | LB_UNet_5s | 0.87 | 0.08 | 0.09 | 14.5 |
| SD (FdN) | 15s | FPT | 0.50 | 0.75 | 0.22 | 7.5 |
| | | HMM | 0.84 | 1.86 | 0.81 | 3.4 |
| | | SV_FCNet_15s | 0.63 | 1.17 | 0.71 | 4.8 |
| | | **SV_CNNet_15s** | **0.87** | **0.59** | **0.83** | **3.8** |
| | | SV_UNet_15s | 0.73 | 0.58 | 0.55 | 5.8 |
| | | LB_FCNet_15s | 0.56 | 0.61 | 0.48 | 6.5 |
| | | LB_CNNet_15s | 0.62 | 0.27 | 0.08 | 12.9 |
| | | LB_UNet_15s | 0.63 | 0.18 | 0.08 | 13.2 |

AUC is for area under the roc curve. BCE is the binary cross entropy. MSE corresponds to the mean square error of the diving distribution maps estimated with kernel density estimations and plotted in Fig 5 to the correct diving distribution. SV is for Peruvian boobies (*Sula variegata*)

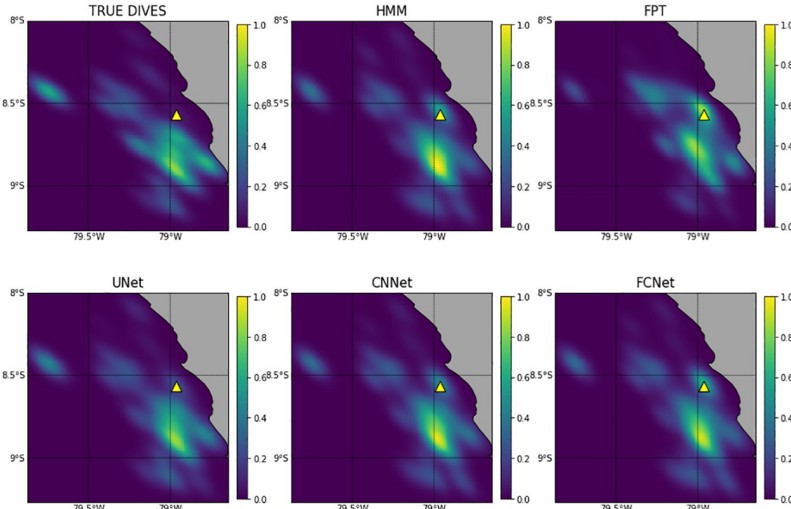

**Fig 6. Maps of dive distributions of Peruvian boobies from Guañape Island.** —These density maps were obtained through Kernel Density Estimation. The top left map has been computed from true dives derived from TDR data. The five other maps are estimations of this map, using all points of the trajectories with weights associated to diving probabilities estimated by the studied approaches: First-Passage Time (FPT), Hidden Markov Model (HMM), Fully Connected Network used by [17] FCNet (top right), fully-Convolutional Network (CNNet), and the U-Network (UNet). Dive map mean square error (MSE) between estimated and reference distribution are in Table 3. Land-sea mask is extracted from GSHHG data (https://www.soest.hawaii.edu/pwessel/gshhg/).

outperformed HMM that predicted around 85% of dives. In particular, the proposed U-shape deep network (UNet) demonstrated a greater robustness to different data inputs, as it obtained the best results whatever the sampling resolution (Table 3).

Additionally, UNet also resulted in better seabird dive distribution maps (Fig 6). Recently numerous studies used seabirds dive as a proxy for prey distribution, and such distribution are usually computed by applying KDE on dive predictions derived from HMMs [55–57]. Here, we show that the error in the estimation of dive distributions maps can be divided by two when using deep learning tools rather than HMM tools. In our specific study, HMMs over-estimated the frequency of dives at specific locations (including the vicinity of the colony). Sulids and cormorants spend time bathing near their breeding territories involving vigorous

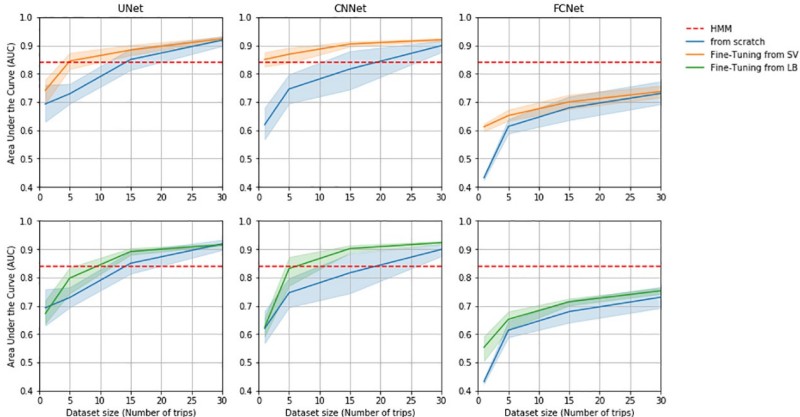

**Fig 7. Fine-tuning.** AUC indices of trained deep network function of dataset size.

splashing and beating the water with the wings [53]. Such behaviours associated to low speed might be erroneously classified as diving behaviour by state-of-the-art tools which could explain the observed bias. This might also explain why HMM are better at predicting boobies' than cormorants' dives because these birds spend more time resting at the surface, which corresponds to low speed patterns without being dives (see Table 1). We may also stress that Cormorants trajectories are characterized by relatively long gaps in the regularly sampled sequence of locations, since these devices do not receive a satellite signal while submerged [26, 58]. This may in turn make more complex the analysis of Cormorants trajectories. In this respect, UNet showed a greater ability to discriminate the resting/bathing behaviours from dives, and a greater robustness to the presence of linearly-interpolated segments. Whereas HMM are mostly driven by fine-scale features (w.r.t. the considered time resolution), UNets exploit a multi-scale analysis of trajectory data and can extract relevant multi-scale information to retrieve dive. Future work could investigate further the key features extracted by UNets. As shown in Fig 3, the performance of the deep networks was closely related to the temporal resolution of the sampled dataset. Whereas HMM did not succeed in exploiting higher-resolution data, UNets led to better performance when the resolution increased. This supports a greater ability of UNets both to deal with potential aliasing effects as well as to exploit fine-scale features. With technological advances in sensor technology, ecologists are able to collect larger amount of data than ever before. We might expect GPS with lower consumption and higher resolution in the future. Such an expected trend would make more critical the exploitation of the proposed deep learning approaches to make the most of the collected high-resolution animal trajectories [13, 59, 60].

When considering neural network approaches, training models which may apply beyond the considered training framework is a key feature, generally referred to as the generalization performance of the trained neural networks. Beyond the evaluation of dive prediction performance on a trajectory dataset, which is independent from the training dataset, the question whether a model trained on a given dataset, e.g. for a given species, colony and time period, may apply to other species, colonies and/or time periods, naturally arises as a key question. Numerous studies in the deep learning literature [61, 62] have highlighted that some neural architectures show relevant generalization properties whereas others may not. Here, we evaluated the generalization performance of the three benchmarked deep networks.

Thus we demonstrate the ability of deep networks trained at a colony for one species to also apply to an another colony (of the same ecosystem) for the same species. In our example, Peruvian boobies from Guañape Island did have different foraging strategies from their counterparts from Pescadores island, with trips two times longer and dives slightly longer (see Table 1). However, the UNet reached similar dive prediction performance when applied to Guañape data. This suggests that dive patterns are highly similar between Peruvian boobies from both colonies. We also show the great ability of the CNNet to generalize dive prediction to a seabird of same genus but from a totally distinct ecosystem. When applied to masked boobies trajectories from a Brazilian colony the CNNet trained from Peruvian boobies data obtained an AUC of 0.87 despite the important difference in foraging strategies (Table 3). The same model trained on masked boobies data reached an AUC of 0.93 (Fig 7), suggesting that diving characteristics are slightly different. Masked boobies from the Brazilian colony feed indeed on different preys, and spend way more time resting at the surface (Table 1). As deep networks trained on cormorants unsurprisingly led to less accurate prediction when used to predict boobies dives, we suggest that the CNNet may capture genus-specific features. These results then support the relevance of deep learning schemes as 'ready-to-use' tools which could be used by ecologists to predict seabirds dives on new (small) datasets, including when these datasets do not include groundtruthed dive data for a supervised training. To make easier such

applications, we share online the different models we trained on the considered datasets (https://github.com/AmedeeRoy/BirdDL/models).

Beyond such a direct application, trained models are also of key interest to explore transfer learning strategies, which refer to the ability of exploiting some previously trained models to address a new task or dataset rather than training a new model from scratch. We illustrated how fine-tuned CNNet and UNet models could outperform HMM with smaller training datasets. For instance, the fine-tuned CNNet for the prediction of masked boobies' dive was able to converge and outperform HMM with a dataset twice as small as the dataset required to reach same performance without fine-tuning (Fig 7). Such a result was even possible by initializing neural networks with the model trained with cormorant data. This further supports the ability of deep networks to generalize their prediction from deep diving seabirds (e.g. cormorants) to plunge divers (e.g. boobies). Fine-tuning is thus particularly relevant when the training dataset may not be sufficiently large to train a model from scratch. While the need of large dataset is often presented as a drawback for supervized techniques, we demonstrated that relatively small datasets (5-10 foraging trips, i.e. 10k to 20k GPS data) may be enough to fine-tune deep networks and outperform state-of-the-art approach to data segmentation. Thus, we expect that our models will be of interest for future work on seabird trajectory segmentation, as they could be used as initializations for fine-tuning procedures.

## Acknowledgments

We thank all people involved in fieldworks: H. Weimerskirch, K. Delord, C. Barbraud, Y. Tremblay, J. Silva, G. Passuni, C. Boyd, C. Saraux, A. Brunel, J.Jacoby, L. Figuereido and G. T. Nunes. We thank Proabonos for permission to work on Guañape and Pescadores Islands. We thank the Brazilian Ministry of Environment (ICMBio) and Fernando de Noronha's firemen for the authorization and logistical support to fieldworks in Brazil.

## Author Contributions

**Conceptualization:** Amédée Roy, Sophie Lanco Bertrand, Ronan Fablet.

**Formal analysis:** Amédée Roy.

**Supervision:** Sophie Lanco Bertrand, Ronan Fablet.

**Writing – original draft:** Amédée Roy.

**Writing – review & editing:** Sophie Lanco Bertrand, Ronan Fablet.

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
