## [Decision Letter · Decision Letter 0]

8 Sep 2021

Dear Mr. Roy,

Thank you very much for submitting your manuscript "Deep learning and Trajectory Representation for the Prediction of Seabird Diving Behaviour" for consideration at PLOS Computational Biology.

As with all papers reviewed by the journal, your manuscript was reviewed by members of the editorial board and by several independent reviewers. In light of the reviews (below this email), we would like to invite the resubmission of a significantly-revised version that takes into account the reviewers' comments.

The reviewers recommendation was between "Major revisions" and "reject". I would like to give the authors a chance to revise, but please try to provide a more in depth investigation of the advantages of DME, and more testing, to demonstrate its advantage and make it more available for usage by others.

We cannot make any decision about publication until we have seen the revised manuscript and your response to the reviewers' comments. Your revised manuscript is also likely to be sent to reviewers for further evaluation.

Sincerely,

Natalia L. Komarova

Deputy Editor

PLOS Computational Biology

Natalia Komarova

Deputy Editor

PLOS Computational Biology

The reviewers recommendation was between "Major revisions" and "reject". I would like to give the authors a chance to revise, but please try to provide a more in depth investigation of the advantages of DME, and more testing, to demonstrate its advantage and make it more available for usage by others.

Reviewer's Responses to Questions

**Comments to the Authors:**

Reviewer #1: Intro.

L.19 & 27 The term “megafauna” is confusing and misused since it is commonly only for large (in size) fauna, such as large herbivores, whales etc. Seabirds are definitely not megafauna. A better term for your study would be top marine predators.

L. 27 To be good indicators they need to be both sensitive and changing in a predictable manner.

L. 32-34 I certainly do not agree with the statement of these 2 sentences. At-sea predation do exist and by-catch as well; they are many papers showing how fishing vessels can be tracked by seabirds and also be negatively impacted (by-catch and easy food). In addition, since they are breeding they foraging is done to secure prey for their brood and this is clearly a constraint on their movements; for instance most penguins do long range movements for their chicks and short and local movements at-sea for their own needs. Please correct the parag accordingly.

MM

L. 87. The authors mentioned that they interpolated missing data. How many were missing? Testing the impact of interpolation on the final analyses is advised.

L.93 Although the splitting % is fine and quite standard for deep learning, it does not give much replicates, which is critical for these data hungry approaches. Seventy % of 234 foraging trips, is 163.8, 20% is 46.8 and 10% is 23.4. Since sample size limit is a function of the complexity of your model (and yours is certainly one), it would be appropriate to quantify the performance of your DL algorithm in response to the amount of data (many models with similar complexity require > 1000 of replicates and to me, the foraging trips are the sample size as the location points within them are not independent). Not only this will help show the readers that your approach is robust, but it will serve as a benchmark for others in the future who may have a different number of foraging bouts. Often in ecological studies, researchers do not have the means to equip that number of individuals so this can help having other using your approach in the future with their own (limited) dataset.

Discussion

A dedicated parag on the ecological aspects of the datasets and the consequences and potential applications for other types of data is warranted.

Editorial issues:

The numbering of the references is all wrong; they are not cited in order; e.g. you start by citing ref #2 L.20 then ref [20) L.22 etc.

Reviewer #2: This study proposes a new neural network model for detecting diving activities.

In my understanding, introducing the distance matrix encoder (DME) is new and successfully improved the detection accuracy.

However, in the current manuscript, I think the advantage of the DME is not fully investigated.

The input of UNet (Fig.3) is time-series of longitude, latitude, and coverage. However, the longitude and latitude are meaningless to detect diving events. To recognize events of moving objects, speed and bearing (angle) are usually used. I consider that when the authors simply use time-series of speed, bearing, and coverage as the input of UNet, the method can achieve good performance comparable to DME-UNet.

I'm also afraid that the contribution of DME is limited because, as shown in the right graph of Fig.3, the performances of DME-UNet and UNet are similar. Can you make this graph using the cormorant data?

Here are my comments that can be beneficial to improving the quality of the paper.

1. It is good to investigate the contribution of DME deeply. As mentioned above, please use speed and bearing speed (radian per time unit) as additional inputs of UNet. Please also make a graph like Fig 3 using the other test data sets.

2. The authors try to detect diving events using only GPS data (without using water depth sensor and accelerometer). However, the motivation is not described in the introduction section.

3. Line 84: How to extract foraging trips from GPS records? Please explain.

4. Line 85: How did you synchronize time stamps of GPS and TDR?

5. Gaps in Table 1 is not explained.

6. The authors use AUC to evaluate the methods. However, the goal of the authors is detect diving events. So, it is better to show the classification performance of the proposed method (e.g., F1-score of diving).

**Have the authors made all data and (if applicable) computational code underlying the findings in their manuscript fully available?**

Reviewer #1: None

Reviewer #2: Yes

PLOS authors have the option to publish the peer review history of their article (what does this mean?). If published, this will include your full peer review and any attached files.

Reviewer #1: No

Reviewer #2: **Yes: **Takuya Maekawa
---

## [Decision Letter · Decision Letter 1]

2 Feb 2022

Dear Mr. Roy,

We are pleased to inform you that your manuscript 'Deep inference of seabird dives from GPS-only records: performance and generalization properties' has been provisionally accepted for publication in PLOS Computational Biology.

Best regards,

Natalia Komarova

Deputy Editor

PLOS Computational Biology

Reviewer's Responses to Questions

**Comments to the Authors:**

Reviewer #2: I think the authors addressed all the concerns I mentioned in the previous version.

I have few minor comments regarding transfer learning.

- In intro, "transfer learning" suddenly appears. It is better to add descriptions regarding the importance of generalization, i.e., motivation of generalization, in the introduction section.

- I think learning rates used were different between training-from-scratch and fine-tuning. It is better to show these information.

**Have the authors made all data and (if applicable) computational code underlying the findings in their manuscript fully available?**

Reviewer #2: None

PLOS authors have the option to publish the peer review history of their article (what does this mean?). If published, this will include your full peer review and any attached files.

Reviewer #2: **Yes: **Takuya Maekawa

---

## [Editor Report · Acceptance letter]

8 Mar 2022

PCOMPBIOL-D-21-00564R1 

Deep inference of seabird dives from GPS-only records: performance and generalization properties

Dear Dr Roy,

I am pleased to inform you that your manuscript has been formally accepted for publication in PLOS Computational Biology. Your manuscript is now with our production department and you will be notified of the publication date in due course.

With kind regards,

Katalin Szabo
